# Spatial Accessibility Assessment of Prehospital EMS with a Focus on the Elderly Population: A Case Study in Ningbo, China

**DOI:** 10.3390/ijerph18199964

**Published:** 2021-09-22

**Authors:** Huanhuan Zhu, Lin Pan, Yiji Li, Huiming Jin, Qian Wang, Xin Liu, Cong Wang, Peng Liao, Xinyang Jiang, Luo Li

**Affiliations:** 1School of Public Health, Fudan University, Shanghai 200032, China; 18621600151@163.com (H.Z.); 20111020057@fudan.edu.cn (L.P.); wangqian0519@126.com (Q.W.); 19211020044@fudan.edu.cn (X.L.); 19111020004@fudan.edu.cn (C.W.); 20111020054@fudan.edu.cn (P.L.); 20111020051@fudan.edu.cn (X.J.); 2Shanghai Institute of Infectious Disease and Biosecurity, School of Public Health, Fudan University, Shanghai 200032, China; 3Ningbo Medical Emergency Center, Ningbo 315000, China; yiji_li@163.com (Y.L.); huiming_jin@163.com (H.J.); 4Key Laboratory of Public Health Safety of the Ministry of Education and Key Laboratory of Health Technology Assessment of the Ministry of Health, Fudan University, Shanghai 200032, China

**Keywords:** prehospital emergency medical services (EMS), spatial accessibility, the elderly, gravity model, empirical Bayesian Kriging (EBK) interpolation analysis

## Abstract

The spatial accessibility of prehospital EMS is particularly important for the elderly population’s physiological functions. Due to the recent expansion of aging populations all over the globe, elderly people’s spatial accessibility to prehospital EMS presents a serious challenge. An efficient strategy to address this issue involves using geographic information systems (GIS)-based tools to evaluate the spatial accessibility in conjunction with the spatial distribution of aging people, available road networks, and prehospital EMS facilities. This study employed gravity model and empirical Bayesian Kriging (EBK) interpolation analysis to evaluate the elderly’s spatial access to prehospital EMS in Ningbo, China. In our study, we aimed to solve the following specific research questions: In the study area, “what are the characteristics of the prehospital EMS demand of the elderly?” “Do the elderly have equal and convenient spatial access to prehospital EMS?” and “How can we satisfy the prehospital EMS demand of an aging population, improve their spatial access to prehospital EMS, and then ensure their quality of life?” The results showed that 37.44% of patients admitted to prehospital EMS in 2020 were 65 years and older. The rate of utilization of ambulance services by the elderly was 27.39 per 1000 elderly residents. Ambulance use by the elderly was the highest in the winter months and the lowest in the spring months (25.90% vs. 22.38%). As for the disease spectrum, the main disease was found to be trauma and intoxication (23.70%). The mean accessibility score was only 1.43 and nearly 70% of demand points had scored lower than 1. The elderly’s spatial accessibility to prehospital EMS had a central-outward gradient decreasing trend from the central region to the southeast and southwest of the study area. Our proposed methodology and its spatial equilibrium results could be taken as a benchmark of prehospital care capacity and help inform authorities’ efforts to develop efficient, aging-focused spatial accessibility plans.

## 1. Introduction

Prehospital emergency medical services (EMS) are an integral part of emergency medical care, which provides basic life support and critical care transportation to the patients. The main aim of this service is to avoid worsening the health condition of the patients on the way to the hospital [1,2].

Studies found that the majority of total prehospital EMS users are elderly patients, and elderly patients presenting higher severity problems appear to be the key driver to the rising demand of prehospital EMS [3,4,5]. It has been proven by a number of studies that 20–55% of the elderly were transported to the emergency departments by ambulances [4,6,7] and the percentage was found to be 2.0–4.4 times higher than other younger age groups [8,9]. According to the United Nations’ definition, “the elderly” refers to the population of those 60 years old and above. When the population aged 60 years and above accounts for 10% or higher of the total population in a country or region, or the proportion of population aged 65 years and above is 7% or higher, it indicates that this country or region becomes an aging society [10]. Nowadays, the proportion of elderly people is steadily increasing globally. About 20% [11], 20.4% [12], 16.6% [13], 28.4% [14], 16.4% [15], and 15.9% [16] of the population in the United Kingdom, France, United States, Japan, South Korea, and Australia are aged 65 years and above, respectively. With the inevitable trend of aging, the mortality pattern has changed from young people to the more complex diseases of old age, and the demand for prehospital EMS has shown a sustainable rise worldwide over the recent years [17]. As one of the largest developing countries, China’s population is aging rapidly due to the one-child policy and the reduced mortality rate. The registered elderly population (aged 65 and above) accounts for 13.50% of the total population (1.41 billion) by 2020, which is a 4.63% increase over the proportion by 2010 [18]. Therefore, China is also challenged to meet the increasing prehospital EMS needs of the elderly [19,20].

Spatial accessibility is defined as the volume and proximity of services provided to the population of interest or the services that are available to a certain region or population given the prevailing transportation system, which mainly considers location and travel impedance [21]. Longer travel distances to healthcare facilities impair accessibility and are related to negative health outcomes, especially in time-sensitive diseases, such as traumatic injuries and heart attacks [22,23], which denote the significance of prehospital EMS allocation’s rationality and the importance of high spatial accessibility to prehospital EMS for the elderly [17,24,25].

There are lots of studies in the literature showing high interest in measuring spatial accessibility to different public services, such as libraries [26], supermarkets [27], urban parks [28], and healthcare facilities [29,30]. However, among the studies of spatial access to healthcare facilities, little attention has been given to the spatial accessibility assessment of prehospital EMS. While the literature abounds with studies on the age-associated patterns of healthcare use and expenditure [31], aging population-focused studies on prehospital EMS are scarce. In China, currently, optimization of the allocation of ambulance bases has not been paid enough attention to, and allocation policies are generally empirical, which are related to the inequity and imbalance of the accessibility of prehospital EMS and are unfavorable for the timely rescue for the elderly [32]. From this standpoint, it is crucial to evaluate and ensure the elderly’s spatial accessibility to prehospital EMS according to the demand characteristics.

As for measuring spatial accessibility to healthcare services, various measures have been proposed, including regional availability model [33], kernel density model [29], gravity model [34], and two-steps floating catchment area (2SFCA) model [30]. The gravity model considers interaction between supply and demand located in different areas, and it can accurately reflect the residents’ access to healthcare services in a small spatial unit, which has been applied in studying healthcare access and other areas. In existing studies, when selecting the research unit to evaluate the accessibility of healthcare facilities, some studies take the county or district as the research scale [35,36] while other studies take the community or town [21]. In fact, there are many situations of cross-district medical treatment in one city. Taking only a certain municipal area as the research scale, the results could be fragmented and insufficient to reflect the overall characteristics of the city. In order to better explore and display the differences in the accessibility of healthcare facilities within a region, the evaluation should be analyzed on a smaller research scale as far as possible, especially in large and medium-sized cities. Besides, residents should be able to get access to healthcare services including prehospital EMS conveniently and impartially. The equilibrium of healthcare services is the fairness level of residents’ access to healthcare services in different locations, which is affected significantly by special accessibility [37]. Therefore, the equilibrium evaluation of spatial access is also crucial. Through equilibrium evaluation results, we can learn the degree of regional equalization and the similarity of accessibility between adjacent demand points. One of the widely used approaches of measuring equilibrium is empirical Bayesian Kriging (EBK) interpolation analysis [38].

In this study, taking Ningbo, China, as an example, we focused on the elderly’s prehospital EMS demand characteristics and evaluated the elderly’s spatial access to prehospital EMS by using the gravity model and EBK analysis. We sought to answer the following specific research questions: In the study area, “what are the characteristics of the prehospital EMS demand of the elderly?” “Do the elderly have equal and convenient spatial access to prehospital EMS?” and “How can we satisfy the prehospital EMS demand of an aging population, improve the elderly’s spatial access to prehospital EMS, and then ensure their quality of life?” In addition, we aimed to help prehospital EMS policymakers to develop efficient aging-focused prehospital EMS accessibility plans by the proposed methodology and case study.

## 2. Materials and Methods

### 2.1. Study Area and Study Population

Ningbo is one of the largest and most populous cities in eastern China with 10 districts and counties. Moreover, it is also facing a severe challenge with regard to the aging population. The registered elderly population (aged 65 and above) accounted for 12.59% of the total population in 2020, which is a 3.98% increase over the proportion in 2010 [39]. The study area of this paper consists of three urban districts of Ningbo, including Haishu District, Yinzhou District, and Jiangbei District, as shown in Figure 1. Moreover, the proportions of the elderly in all these three districts are higher than 10% [40,41,42]. The specific information is summarized in Table 1. In addition, the elderly aged 65 and above who used the ambulance in 2020 in the study area are chosen as our study population.

Prehospital EMS care in China, including Ningbo, is administered and organized as a public service by the Ministry of Health. In Ningbo, anyone can use ambulance services by calling 120 and can get access to prehospital EMS by the dispatch operators at dispatch center in Ningbo Medical Emergency Center. Prehospital EMS crew members are physicians, emergency medical technicians, and ambulance drivers in Ningbo.

Recently, Coronavirus disease 2019 (COVID-19) has substantially impacted the EMS system. According to the requests of the Transfer Plan for Pneumonia Cases Infected by COVID-19 issued by the National Health Commission, emergency centers should be equipped with specialized medical personnel, drivers, and ambulances to perform the task of transferring pneumonia cases infected by COVID-19 under the requirements of its specific workflow [43]. Therefore, since the first confirmed positive test for severe acute respiratory syndrome coronavirus-2 (SARS-CoV-2) in January 2020, in addition to its daily EMS missions, Ningbo Medical Emergency Center has also undertaken this special transfer mission.

### 2.2. Data Collection and Sources

The data used in this study included supply side, demand side, and administration boundary data along with road network. The demand data consisted of elderly population data and emergency call data. The supply data mainly referred to the data of ambulance bases.

#### 2.2.1. Elderly Population Data

The elderly population data were collected from the bulletins of the 7th National Census that were published on official websites. When calculating prehospital EMS demand rates (per 1000 population), we used the collected population numbers.

#### 2.2.2. Elderly’s Emergency Call Data

Emergency data were acquired from Ningbo Medical Emergency Center, which covered the period between 1 January 2020, to 31 December 2020. Each emergency call record included the information about the patient’s age, gender, call date, pick-up address (in the form of longitude and latitude), chief complaint, preclinical diagnosis, disease category and ring time, pick-up time, dispatch time, departure time, arrival time, and other key time nodes’ explicit time.

#### 2.2.3. Ambulance Bases Data

The ambulance bases data also came from Ningbo Medical Emergency Center, including ambulance base name, location (in the form of longitude and latitude), and the number of ambulances. No scientific criteria were used to locate the bases, and they were generally in or adjacent to one comprehensive hospital. There were 4 ambulance bases in Jiangbei District, 8 in Yinzhou District, and 6 in Haishu District. According to the interview with the director of the dispatching section of Ningbo Medical Emergency Center, each ambulance base was equipped with 2 ambulances.

#### 2.2.4. Administration Boundary Data

Administration boundary data was collected from the Topographic Database of the National Fundamental Geographic Information System of China, which was provided by the National Geomatics Center of China. The study area contains 49 counties, towns, and streets (Figure 2A).

#### 2.2.5. Road Network

A road network was used to compute the shortest travel distance from each demand point to the ambulance base. We gathered road network data from the Topographic Database of the National Fundamental Geographic Information System of China, which was provided by the National Geomatics Center of China (Figure 2B).

### 2.3. Methodology

The study was approved by the Ethics Committee of School of Public Health of Fudan University. We conducted the study mainly in three stages as follows.

#### 2.3.1. Stage 1: Data Process

As previously stated, we obtained 2020 EMS call data from Ningbo Medical Emergency Center, consisting of 38,822 cases. Figure 3 summarizes the schematic for data management. We excluded records that patients’ ages were missing, callers’ locations were unavailable, or serial numbers were duplicated.

#### 2.3.2. Stage 2: Measuring Spatial Accessibility

To estimate the elderly’s spatial access to prehospital EMS, we used the gravity model. Our estimation process can be divided into three steps as follows:Step 1: Definition of demand points

In our study, we did not use the counties, towns, communities, or residential areas as the study units. In order to reflect the characteristics of the prehospital EMS needs of the elderly in the study area and make it more integrated and accurate, we divided the study area into a fishnet composed of several 1000 × 1000 m^2^ grids (Figure 4A). Then, we located the total emergency calls of the elderly on the map (Figure 4B), making a spatial connection with the fishnet. By summarizing and counting the number of calls in each grid, we selected the grids whose count numbers were 0, extracted them while the left grids were transformed into points, and we defined these points as the demand points (Figure 4C) that were used in the later calculation.

Step 2: Locating the current ambulance bases

According to the locations in latitude and longitude format of the current ambulance bases, we located them on the map (Figure 5).

Step 3: Evaluation of the spatial accessibility of demand points

As introduced in the previous section, accessibility measurements of public service facilities have been thoroughly discussed in previous studies. As the gravity model can not only consider the potential capability of the demand, supply, and the distance impedance between the two aspects but also accurately reflect the residents’ access to healthcare services in a small spatial unit, we employed it in the present study [34,44].

In our study, this model was used to capture the interactions between ambulance availability and the demand distribution, generating an access score for each demand point by using Equations (1) and (2) as follows:(1)$$A_{i}=\overset{n}{\underset{j=1}{\sum }}\frac{M_{j}}{D_{ij}^{\beta }V_{j}}$$
(2)$$V_{j}=\overset{m}{\underset{k=1}{\sum }}\frac{P_{k}}{D_{kj}^{\beta }}$$
where $A_{i}$ represents the spatial accessibility of the demand point *i* to prehospital EMS ambulance bases.

$M_{j}$ represents the service capability of the ambulance base *j*, and here it was represented by the amount of ambulance base *j*.

$D_{ij}$ is the distance or travel time between *i* and *j*. In this study, we used distance.

$V_{j}$ is the influencing factor of population size.

$P_{k}$ is the amount of emergency calls in demand point *k*.

*β* is the distance-decay parameter. As summarized by Peeters and Thomas, the distance-decay parameter *β* in existing studies lies between 0.9 and 2.29 [45]. Wang and Zhang conducted a sensitivity analysis on *β*, setting *β* as 1 or 2 respectively. The results showed that 2 is better for measuring the accessibility of healthcare facilities [46,47]. Thus, we set *β* as 2 in this study.

*n* and *m* represent the number of ambulance bases and demand points, respectively.

After obtaining the accessibility index $A_{i}$, we classified them quantitatively and made them visualized.

#### 2.3.3. Stage 3: Measuring Spatial Equilibrium of the Whole Study Area

To describe the spatial accessibility of prehospital emergency facilities at known demand points and those in unknown areas around them, it is necessary to analyze the difference of the accessibility calculated by the previous gravity model. The empirical Bayesian Kriging (EBK) interpolation method was used in this study [48]. First, the ArcGis10.2 Geostatistical Analyst module’s normal QQ Chart (Environmental Systems Research Institute, Redlands, California, USA )was utilized to test whether the acquired accessibility scores conformed to normal distribution. Second, we used the Moran’s I index to explore the clustering patterns of the elderly’s spatial accessibility to prehospital EMS. The ArcGis10.2 geoprocessing tool (Environmental Systems Research Institute, Redlands, California, USA) contains an implementation of the Moran I which allows users to identify the spatial agglomeration areas. The significance of spatial autocorrelation is judged by the standardized statistic Z. At a confidence level of 0.05, the absolute value of Z equals 1.96. When the absolute value of Z exceeds 1.96, the spatial autocorrelation is significant. Last, we employed the ArcGis10.2 Geostatistical Analyst module’s geostatistical Wizard (Environmental Systems Research Institute, Redlands, California, USA) to construct the EBK model.

## 3. Results

### 3.1. Basic Characteristics of the Study Elderly Population

Table 2 demonstrates characteristics of calls from 1 January 2020, to 31 September 2020, in the study area, Ningbo. From 10,062 elderly patients who received prehospital EMS care, 53.66% were male. There were 37.01% of patients aged 65–74, 31.67% aged 75–84, 28.33% aged 85–94, 2.96% aged 95–104, and 0.03% aged 105 and over. The mean age was 78.79 years old.

### 3.2. Distribution Characteristics of the Elderly’s Demand for Prehospital EMS

#### 3.2.1. Time Distribution

Ambulance utilization by the elderly was highest in December, whereas it was the lowest in February (10.04% vs. 6.74%). Ambulance use of the elderly was the highest in the winter months (25.90%) and the lowest in the spring months (22.38%). Calling time interval showed peaks at 6:00–8:59 a.m. and 9:00–11:59 a.m. The frequency of calls decreased significantly after midnight (Figure 6).

#### 3.2.2. Disease Distribution

The top five of the 15 illness categories regarding prehospital EMS demand of the 10,062 elderly patients were trauma and intoxication (23.70%), neurologic diseases (14.90%), cardiovascular diseases (11.12%), respiratory diseases (10.23%), and gastrointestinal diseases (8.47%) (Table 3).

#### 3.2.3. Calls’ Spatial Distribution

The total number of the elderly population aged 65 and above in the study area was 367,346, and 10,062 patients were admitted to pre-hospital EMS in 2020. According to the address data (in the format of longitude and latitude data) of 10,062 calls, with the help of ArcGis10.2 software, we located the call points of the elderly in the digital map (Figure 4B). The calling points were randomly spread through the study area. Hence, 3960, 1856, and 4262 elderly people in Haishu District, Jiangbei District, and Yinzhou District made emergency calls and used the prehospital EMS care services. Thus, the rate of ambulance service for the studied elderly people was 27.39 per 1000 elderly residents (10.78 in Haishu district, 5.05 in Jiangbei district, and 11.56 in Yinzhou district) (Table 4). According to the known total number of permanent residents and the number of the elderly population aged 65 and above, we learned that there were 2,772,329 younger people in the study area. Meantime, the younger population’s EMS call number was 16816. Therefore, the rate of ambulance service for the younger people was 6.07 per 1000 younger residents.

#### 3.2.4. Demand Points’ Spatial Distribution

According to the steps of identifying the elderly’s demand points in the methodology part, we defined 793 demand points. Each demand point’s EMS calls amount was counted by ArcToolbox’s spatial connection function. In our study, the calls amount of each demand point was used as $P_{k}$ to calculate the $V_{j}$.

As shown in Figure 7, in the central area, the elderly’s demand for prehospital EMS was relatively higher than those in the surrounding areas. In 2020, among the 793 demand points, the maximum value of EMS calls was 174, where it is in Yinzhou District’s Baihe Street. The minimum value was 1, and there were 184 demand points whose call amount was 1, covering Jiangbei District, Haishu District, and Yinzhou District.

### 3.3. The Elderly’s Spatial Accessibility of Prehospital EMS

Figure 8A shows the elderly’s spatial accessibility scores in different demand points. The closer the color was to dark blue, the lower the accessibility score was. The closer it was to dark red, the higher the accessibility score was. The lowest accessibility score was less than 0.04, which signifies the area where there were less than 0.04 ambulances accessible for one thousand elderly people. The highest accessibility score was higher than 99.66, which means the certain area had more than 99.66 ambulances accessible for one thousand elderly people. The lowest score area and highest score area were displayed in Zhangshui Town of Haishu District and Wuxiang Town of Yinzhou District, respectively.

In addition, in terms of the proportion of accessibility scores, 538 (67.84%) demand points had scored lower than 1, while 239 (30.14%) demand points had scored between 1 and 10, and only 16 (2.02%) higher than 10 (Table 5). The mean accessibility score in study area was 1.43, which means that there were nearly 1.43 ambulances accessible for one thousand elderly people.

### 3.4. Spatial Equilibrium of the Elderly’s Spatial Accessibility to Prehospital EMS

The EBK interpolation analysis here was used to describe the spatial accessibility of prehospital EMS at the demand points along with the surrounding areas and to learn its spatial equilibrium situation. In this study, after logarithmic transformation, the data passed the normality test. By calculation, the Moran’s I index was 0.63 and the standardized statistic Z was 63.07, which denoted that there existed spatial autocorrelation for the accessibility, showing the clustering trend.

As depicted in Figure 8B, on the whole, the spatial accessibility values had remarkable circle characteristics, showing a decreasing trend from the central area to the surrounding area. Except for the central region, there were two areas of higher spatial accessibility values, which were located in the southeast and southwest of the study area. In other words, the elderly’s spatial access to prehospital EMS was imbalanced.

## 4. Discussion

Spatial accessibility of prehospital EMS is particularly important for the physiological functions in the elderly population [49,50]. Due to the recently expanding aging population in the whole world, aging people’s spatial accessibility to prehospital EMS faces a serious challenge. Ningbo, China, also is faced with this very problem. To the authors’ knowledge, most researchers show high interest in measuring the spatial accessibility of various public services, but few researchers have recognized the need to study prehospital EMS accessibility with respect to aging populations. This study aimed to describe the elderly-level access to prehospital EMS and its spatial equilibrium as a benchmark of prehospital care capacity and to inform policymakers of emergency infrastructure reform priorities.

Age is an important factor influencing the demand for prehospital EMS. In our study, approximately 37.44% of patients admitted to prehospital EMS in 2020 were 65 years and older, which was consistent with the data reported in the 2015 National Hospital Ambulatory Medical Care Survey in the US (34.1% of patients 65 years old and above arrived at the emergency departments by ambulance) [51]. Besides, in our study, the rate of utilization of ambulance services by the elderly was 27.39 per 1000 elderly residents, more than 4.51 times the rate for younger residents, which proved that the demand for prehospital EMS was generally much higher among the older population than the younger population. Moreover, Pembe Keskinoglu’s study drew a similar conclusion [9]. Compared to the younger adults, the older adults’ high need for ambulance service might relate to their weaker mobility [52] and cognitive, physical, mental, emotional, and health limitations [53]. These findings confirmed the necessity of prehospital EMS studies to focus on the elderly once again.

As for the distribution characteristics of the elderly’s demand for prehospital EMS, we found that the ambulance utilization rate among the elderly was lower during spring months, which correlated with Ningbo authorities’ coronavirus mitigation strategies during the COVID-19 pandemic outbreak with quarantine precautions, such as the Stay-at-Home order, remote learning for students, and so on [54]. Throughout the day, the utilization rate was observed to decrease at between 0 am and 6 am and to peak at noon. Several studies also reported similar findings [9,55]. In the prehospital EMS disease spectrum of the elderly, the main disease was found to be trauma and intoxication (23.70%). The call rate for infectious and parasitic diseases was very low (0.08%). In view of the above phenomena, we recommend that the current ambulance bases should be equipped with sufficient ambulances to better cope with the peak hours of day and year. In addition, it is necessary to strengthen the skills of prehospital EMS crew for the diagnosis and rescue of common elderly emergencies, especially for trauma management and poisoning rescue, which has been proven to be associated with improved outcomes [56,57,58].

From the perspective of demand points, we observed that the elderly’s spatial access to prehospital EMS was not high. The mean accessibility score was only 1.43 and nearly 70% of demand points had scored lower than 1, which means there were only 1 or even fewer ambulances providing prehospital EMS per 1000 elderly persons. Besides, the gap between the highest score and lowest score was huge (0.04 vs. 99.67). That was attributed to the ambulance bases being concentrated in the central area, and the central region’s road network was more developed, while those of the surrounding new urban developing areas were relatively insufficient.

From the perspective of the whole study area, we found the elderly’s spatial access to prehospital EMS was imbalanced. The global autocorrelation of spatial accessibility values was very strong, showing a high level of spatial clustering. Reflected on the map, the elderly’s spatial accessibility to prehospital EMS had a central-outward gradient decreasing trend from the central region to the southeast and southwest of the study area. That was because the layout of ambulance bases in the southeast and southwest of the study area was almost blank, such as Zhangshui Town, Jiangshan Town, Dongqiao Town, Hengxi Town, Tangxi Town, and Dongqian Lake Street. This implied that, in comparison with the elderly residents who were living in suburbs, elderly residents in central urban areas could obtain prehospital EMS more easily. Previous evaluation studies on the other kinds of medical resources services, several provinces or cities had similar characteristics. China’s Henan province [37], Sichuan province [59], and Shenzhen city [21] all presented resource disparity over space, which was affected by terrain or the medical resources’ layout. Across Ghana, medical resources were also unevenly distributed, with it remaining unreachable in 40.6% of counties to get to the nearest medical services within 1 h [60]. Therefore, access equilibrium’s improvement should be paid more attention to in order to avert death and disability related to unmet prehospital care. For Ningbo, in the area with sufficient resources, which here mean the junction areas of Yinzhou District, Haishu District, and Jiangbei District, its dispatching management process should be further optimized. For areas with scarce resources, increasing ambulance bases capacity or adding additional stations might be the priorities [17,60].

Prehospital EMS and its ambulance stations play an important role in caring for the elderly. The 14th Five-Year Plan for Health Development in Ningbo [61] issued by Ningbo local government has proposed series of action guidelines, such as the establishment of municipal geriatric hospitals and municipal geriatric medical centers. However, the concrete layout plan of prehospital EMS stations to meet the demand of the elderly is not reflected. Through our analysis of the elderly-level access to prehospital EMS, we summarized the characteristics of the elderly’s utilization of prehospital EMS and identified areas with poor spatial accessibility for the elderly, which would be useful for the work to improve prehospital care coverage and ensure the equilibrium between different areas. Compared with previous studies, this study focused on the elderly population, and its study unit was not administrative division or community, but the EMS demand point of the elderly who actually had made the EMS calls. Thus, we were able to take advantage of a fine-scale dataset to produce higher-quality results.

This study has the following limitations: Firstly, since the automatic recording of pre-hospital EMS information of Ningbo during our study period was based on the manual confirmation of an ambulance crew, the documentation of age, gender, time, and address in the course of emergencies was sometimes incomplete. This might have introduced bias in our results. Secondly, because of the incomplete information on disease categories, we classified the missing diseases of elderly patients according to their complaints retrospectively. Thus, inconsistencies might have caused bias regarding the results of the disease spectrum analysis. Thirdly, on-site treatment was not documented, and subsequent in-hospital treatment data were difficult to collect. Therefore, this study did not fully demonstrate the current status of EMS in Ningbo.

## 5. Conclusions

The elderly’s access to prehospital EMS will continue to be a significant challenge, especially in developing countries. It is then crucial for the authorities to employ methods that measure the accessibility of such services in a way that aims for an equilibrium of service provision focusing on the elderly population. In this research, we used the gravity model and EBK interpolation analysis, by defining the demand points and replacing the traditional community points or administrative division points, to measure the elderly’s spatial accessibility to prehospital EMS.

Our finding showed that the elderly’s spatial access to prehospital EMS was imbalanced in the study area, having a central-outward gradient decreasing trend from the central region to the southeast and southwest. In the area with sufficient resources of Ningbo, which here mean the junction areas of Yinzhou District, Haishu District, and Jiangbei District, the dispatching management process should be further optimized. For areas with scarce resources, increasing ambulance bases capacity or adding additional stations might be the priorities. In addition, it is necessary to strengthen the skills of the prehospital EMS crew for the diagnosis and rescue of common elderly emergencies, especially for trauma management and poisoning.

## Figures and Tables

**Figure 1 ijerph-18-09964-f001:**
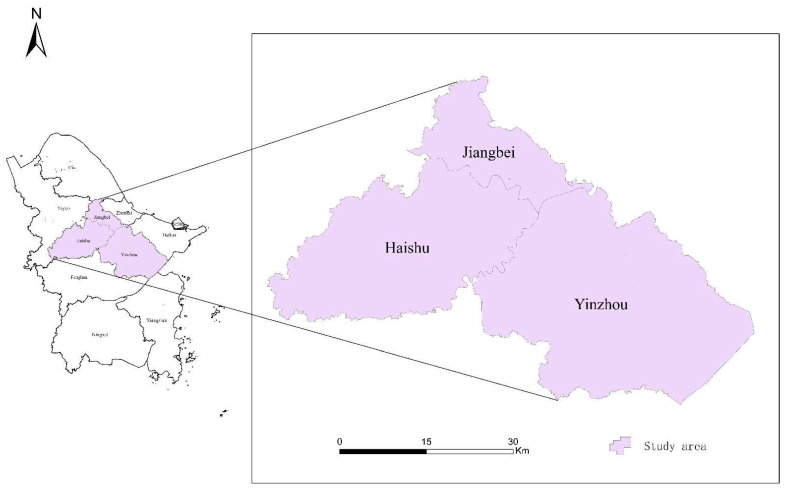
Study area.

**Figure 2 ijerph-18-09964-f002:**
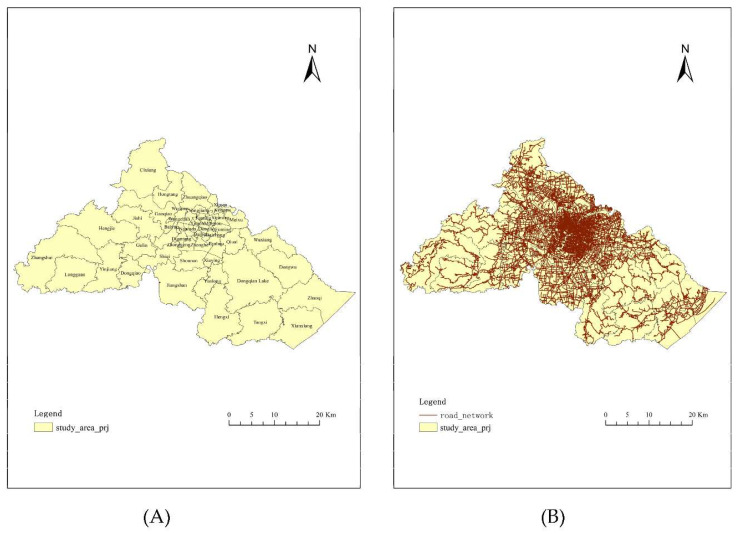
(**A**)The administration boundary; (**B**) Road network.

**Figure 3 ijerph-18-09964-f003:**
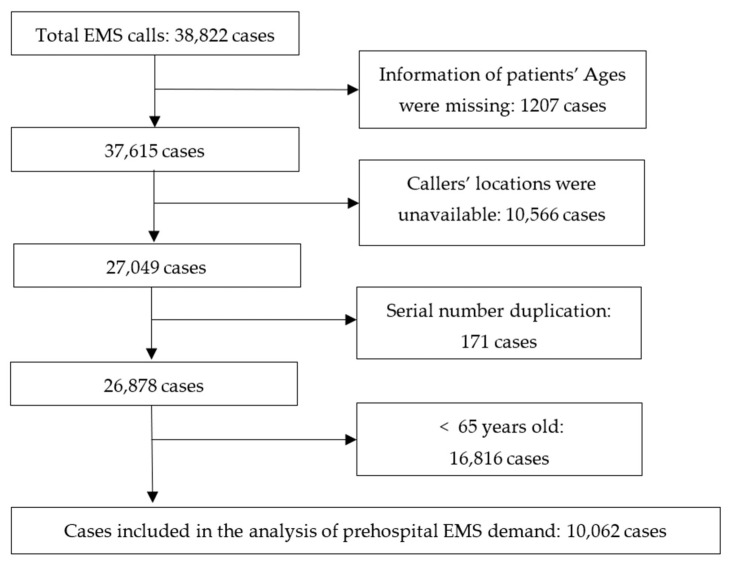
Summary of the schematic of the data management.

**Figure 4 ijerph-18-09964-f004:**
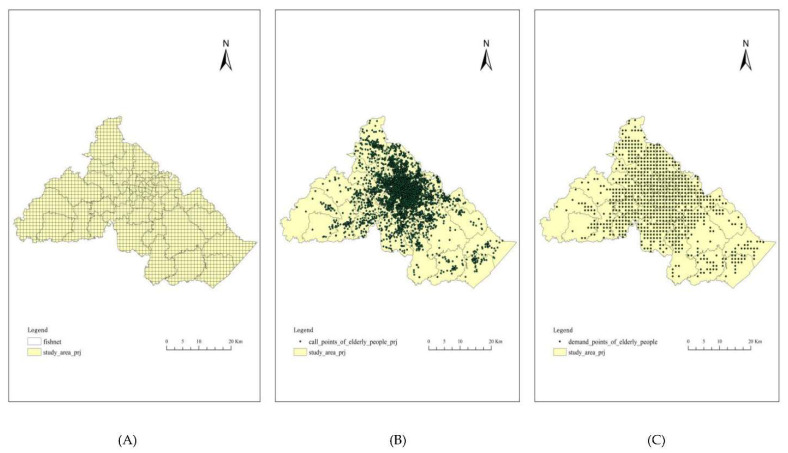
(**A**) Fishnet of study area; (**B**) EMS call points of the elderly; (**C**) Demand points of the elderly.

**Figure 5 ijerph-18-09964-f005:**
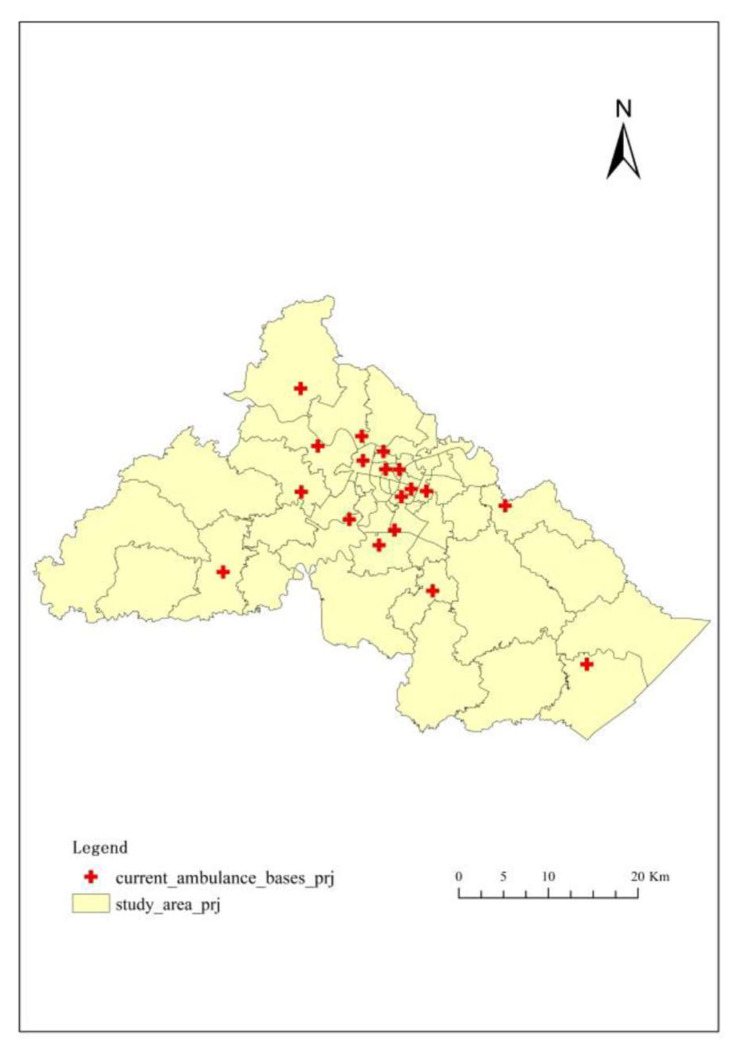
Current ambulance bases of the study area.

**Figure 6 ijerph-18-09964-f006:**
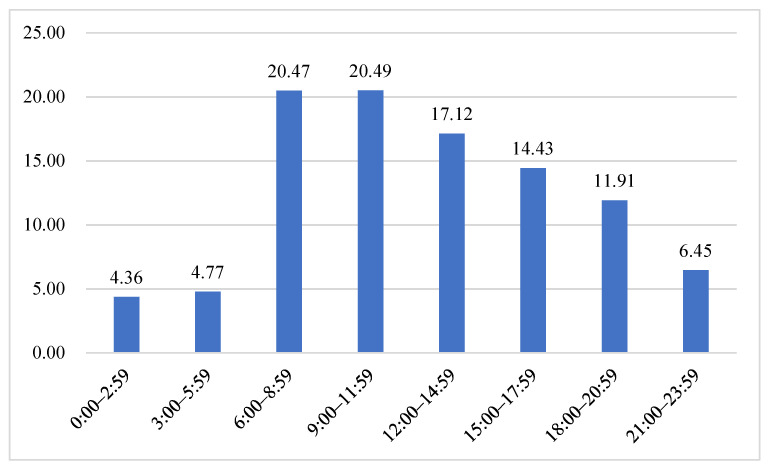
Percentage distribution of emergency calls by hours of the day.

**Figure 7 ijerph-18-09964-f007:**
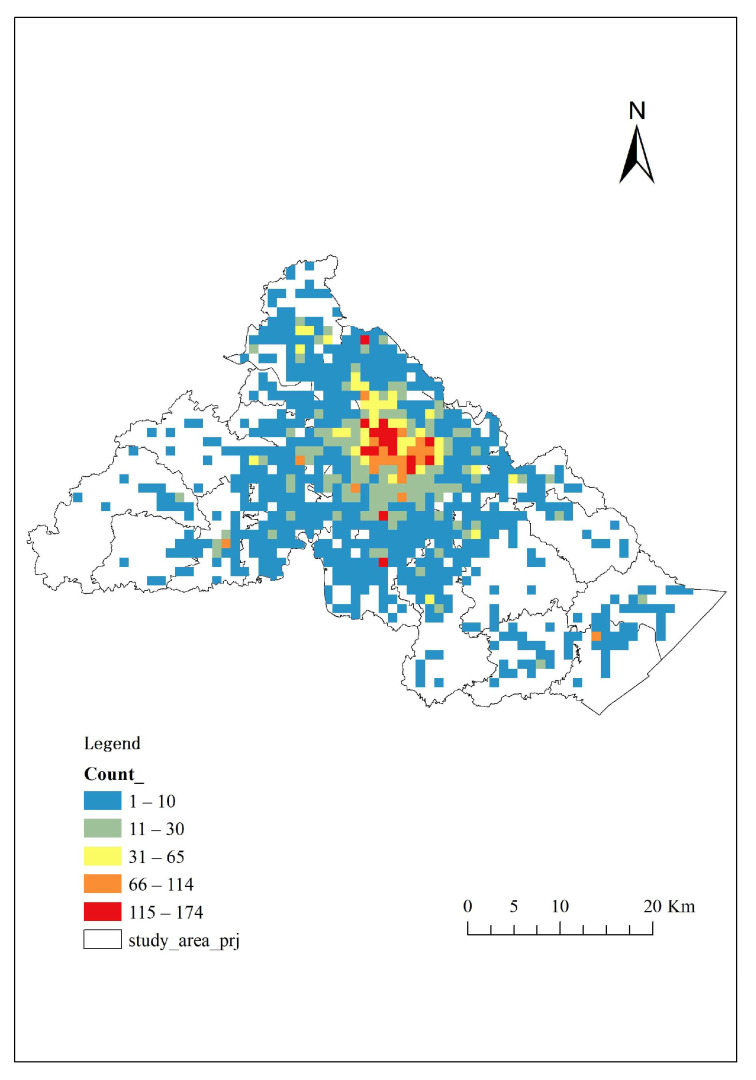
The elderly’s calls amount of demand points.

**Figure 8 ijerph-18-09964-f008:**
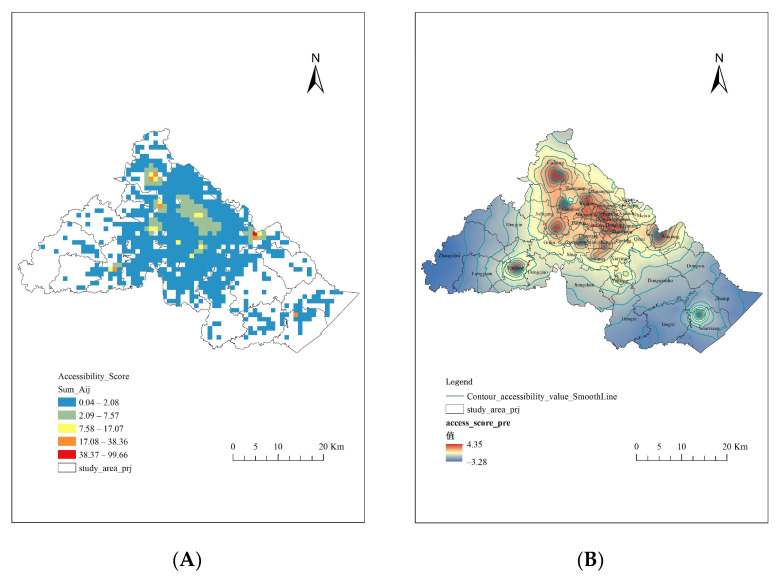
(**A**) The elderly’s spatial accessibility scores of demand points; (**B**) Spatial equilibrium of the elderly’s spatial accessibility in the total study area.

**Table 1 ijerph-18-09964-t001:** Summary of the elderly (aged 65 and above) in Study area, 2020.

Districts	Permanent Residents (*n*)	Aged 65 and Above (*n*)	Percentage of the Elderly (%)
Haishu District	1,041,285	131,201	12.60
Jiangbei District	488,885	57,383	11.74
Yinzhou District	1,609,555	178,762	11.11
Study area	3,139,725	367,346	11.70

**Table 2 ijerph-18-09964-t002:** Basic characteristics of elderly patients.

Characteristics	All Patients (*n* = 10,062)
Sex (%)	
Male	5399 (53.66)
Female	4663 (46.34)
Age (%)	
65–74	3724 (37.01)
75–84	3187 (31.67)
85–94	2851 (28.33)
95–104	297 (2.96)
105 and above	3 (0.03)
Mean age	78.79

**Table 3 ijerph-18-09964-t003:** Ambulance crew diagnosis regarding elderly patients.

Disease Diagnosis	*n*	%
Trauma and intoxication	2385	23.70
Unclassified diseases	1969	19.57
Neurologic diseases	1499	14.90
Cardiovascular diseases	1119	11.12
Respiratory diseases	1029	10.23
Gastrointestinal diseases	852	8.47
Muscle, skeletal and connective tissue diseases	341	3.39
Others	263	2.61
Urogenital diseases	217	2.16
Tumor and blood-related diseases	154	1.53
Endocrine nutritional metabolic diseases	135	1.34
Skin and subcutaneous tissue diseases	41	0.41
ENT diseases	35	0.35
Psychiatric illness	15	0.15
Infectious and parasitic disease	8	0.08

**Table 4 ijerph-18-09964-t004:** The ambulance utilization by the elderly in 2020.

Districts	Permanent Residents	Aged 65 and Above	Ambulance Utilization by the Elderly	Number of Ambulances Utilization per 1000 Elderly Residents
Haishu District	1,041,285	131,201	3960	10.78
Jiangbei District	488,885	57,383	1856	5.05
Yinzhou District	1,609,555	178,762	4246	11.56
Study area	3,139,725	367,346	10,062	27.39

**Table 5 ijerph-18-09964-t005:** Accessibility scores of different demand points.

Accessibility Score	0~1	1~10	10~20	20 and Over
Number of demand points	538	239	12	4
Proportion (%)	67.84	30.14	1.51	0.51

## Data Availability

Our study data are mainly second-hand data. Further inquiries can be directed to the corresponding authors.

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
