# Peer review of "Spatial Accessibility Assessment of Prehospital EMS with a Focus on the Elderly Population: A Case Study in Ningbo, China"

_ijerph, 2021, doi:10.3390/ijerph18199964_

Round 1
Reviewer 1 Report
The study evaluated the spatial accessibility of EMS in a region of China. The research question is important, and the rationale for the study is well presented.
However, I feel utterly surprising the authors do not mention the COVID-19 pandemic despite the study period being January to December 2020, the peak of the COVID-19 pandemic in China. The covid-19 pandemic and mitigation strategies affected the EMS workflow in a very significant and unprecedented ways, so it is impossible to draw conclusions without evaluating this aspect. It would also explain the lower utilization of EMS in the spring months.
The authors should clarify if Figures 7 and 8 refer to elderly patients calling points or the whole population.
On other note to improve the manuscript:
- It would be informative to compare the proportion of the elderly population in China to other countries in the region, Europe or the USA.
- Figure 3: does not tell the sample size for “Aged 65 or over”.
- the manuscript is written in a difficult to read style. I would suggest reducing the number of sophisticated and unconventional terms to improve readability.
. Compare study accessibility scores to scores in other countries or regions.
Reviewer 2 Report
Major comments
- I could understand that this study measures the spatial accessibility to prehospital EMS, but specific research questions are unclear. It would be more straightforward if you could provide specific research questions.
- In addition, what does it mean by equilibrium? Do you mean inequity?
- Population data may be varying over time. I am just wondering how you manipulate the population data. Is the time period of the population data the same to the emergency data (January 1, 2020 to December 31, 2020)?
- Could you provide the reference for the spatial accessibility measurement? (Page 7).
- Compared to the method explanation about the spatial equilibrium, the results are presented very simple. I am wondering Moran’s I and other statistics. You also mentioned the spatial autocorrelation in the discussion section, which means you have to provide the results in advance.
Minor comments
- I do not think you need to display Figure 7 (A) or (B). This represents the same stuff. I also it would be better to change the title of Figure 7. In the figure, you do not have to say “visualized in ArcMap” and “visualized in ArcScene”. 2D or 3D map sound more proper.
- Actually, 4 ~ 6 groups are more appropriate in the map. You may wish to change the number of graphs in the choropleth maps in the manuscript.
- Some figures include Chinese characters.
Round 2
Reviewer 1 Report
I thank the authors for sending the revised manuscript and for studying this important topic. The comments from the previous review were adequately addressed.
Minor additional comments:
You could state your main findings in the end of the abstract to make your results more visible.
Please state the definition of the elderly population in the beginning of the introduction before describing the proportion of older people living in different countries.
Figure 3: I believe the term “Aged 65 and above” should be instead “<65 years old”. I assume that the patients that were included were ≥65 years old, not younger than 65 y.o. Now it seems that patients ≥65 y.o. were excluded.
Reviewer 2 Report
Thank you for your revision.
I've enjoyed reading your revised paper.
Author Response
Dear reviewer,
We deeply appreciate your review. Under your constructive guidance, our manuscript is more reasonable and scientific in terms of structure, visual expression and so on. Thanks again!
Best regards,
Li Luo.